# Retroviral integration into nucleosomes through DNA looping and sliding along the histone octamer

Marcus D. Wilson [1,7,10], Ludovic Renault[1,8,10], Daniel P. Maskell [2,9], Mohamed Ghoneim[3,4], Valerie E. Pye [2], Andrea Nans[5], David S. Rueda [3,4], Peter Cherepanov[2,6] & Alessandro Costa [1]

Retroviral integrase can efficiently utilise nucleosomes for insertion of the reverse-transcribed viral DNA. In face of the structural constraints imposed by the nucleosomal structure, integrase gains access to the scissile phosphodiester bonds by lifting DNA off the histone octamer at the site of integration. To clarify the mechanism of DNA looping by integrase, we determined a 3.9 Å resolution structure of the prototype foamy virus intasome engaged with a nucleosome core particle. The structural data along with complementary single-molecule Förster resonance energy transfer measurements reveal twisting and sliding of the nucleosomal DNA arm proximal to the integration site. Sliding the nucleosomal DNA by approximately two base pairs along the histone octamer accommodates the necessary DNA lifting from the histone H2A-H2B subunits to allow engagement with the intasome. Thus, retroviral integration into nucleosomes involves the looping-and-sliding mechanism for nucleosomal DNA repositioning, bearing unexpected similarities to chromatin remodelers.

[1] Macromolecular Machines Laboratory, The Francis Crick Institute, NW1 1AT London, UK. [2] Chromatin structure and mobile DNA Laboratory, The Francis Crick Institute, London NW1 1AT, UK. [3] Single Molecule Imaging Group, MRC London Institute for Medical Science, London W12 0NN, UK. [4] Molecular Virology, Department of Medicine, Imperial College London, London W12 0NN, UK. [5] Structural Biology Science Technology Platform, The Francis Crick Institute, London NW1 1AT, UK. [6] Department of Medicine, Imperial College London, St-Mary's Campus, Norfolk Place, London W2 1PG, UK. [7] Present address: Wellcome Centre for Cell Biology, University of Edinburgh, Edinburgh EH9 3JR, UK. [8] Present address: NeCEN, University of Leiden, 2333CC Leiden, Netherlands. [9] Present address: Faculty of Biological Sciences, Leeds LS2 9JT, UK. [10] These authors contributed equally: Marcus D. Wilson, Ludovic Renault. Correspondence and requests for materials should be addressed to D.S.R. (email: david.rueda@imperial.ac.uk) or to P.C. (email: peter.cherepanov@crick.ac.uk) or to A.C. (email: alessandro.costa@crick.ac.uk)

Integration of the reverse-transcribed retroviral genome into a host-cell chromosome is catalysed by integrase (IN), an essential viral enzyme (reviewed in[1]). To carry out its function, a multimer of IN assembles on viral DNA (vDNA) ends forming a highly stable nucleoprotein complex, known as the intasome[2–4]. In its first catalytic step, IN resects 3′ ends of the vDNA downstream of the invariant CA dinucleotides (3′-processing reaction). It then utilises the freshly released 3′-hydroxyl groups as nucleophiles to attack a pair of phosphodiester bonds on opposing strands of chromosomal DNA, cleaving host DNA and simultaneously joining it to 3′ vDNA ends (strand transfer reaction)[5,6].

Many important questions pertaining to the nature of the host-virus transactions on chromatin remain unanswered. In particular, it is unclear what role chromatin structure plays in the integration process. Strikingly, although only a fraction of the nucleosomal DNA surface is exposed within the nucleosome core particle (NCP)[7–9], nucleosomal DNA packing does not impede and rather stimulates integration[10–15]. Because retroviral INs have long been known to prefer bent or distorted targets, bending of DNA as it wraps around the histone octamer was thought to facilitate integration into NCPs[12,13]. However, recent structural data revealed that retroviral intasomes require target DNA to adopt a considerably sharper deformation than the smooth bend observed on NCPs[15–19].

Intasome structures from several retroviral genera have been determined by X-ray crystallography and cryo-EM[4,17–20]. Despite considerable variability, all intasomes were found to contain the structurally conserved intasome core assembly minimally comprising four IN subunits synapsing a pair of vDNA ends. Depending on the retroviral species, the core assembly can be decorated by a number of additional IN subunits. The nucleoprotein complex from the prototype foamy virus (PFV) contains only a tetramer of IN, making this well-characterised intasome an ideal model to study the basic mechanisms involved in retroviral integration. Recently, we reported a cryo-EM structure of the pre-catalytic PFV intasome engaged with an NCP at 7.8 Å resolution[15]. Despite the modest level of detail, the cryo-EM data revealed that intasome induces the sharp bending of the nucleosomal DNA by lifting it off the face of the histone octamer at the site of integration. In doing so, the intasome makes supporting interactions with the H2A-H2B heterodimer and the second gyre of the nucleosomal DNA[15]. Due to the limited resolution of the original structure, it was impossible to visualise the conformational rearrangements in the nucleosomal DNA that lead to its disengagement from the nucleosomal core at the site of integration. Thus, it remains to be established whether nucleosomal DNA deformation at the integration site is merely accommodated by local deformation of the duplex DNA structure, or it rather involves global repositioning of the nucleosomal DNA along the histone octamer. In addition, a systematic analysis is needed to understand potential role of histone tails in intasome engagement.

Herein, we employ a combination of cryo-EM and single-molecule Förster resonance energy transfer (FRET) assay to understand what impact retroviral integration has on the structure of the target NCP. We find that strand transfer causes both nucleosomal DNA looping, as well as sliding by two base pairs along the histone octamer. With our findings we uncover unexpected similarities between the mechanisms of retroviral integration and ATP-dependent chromatin remodelling[21–23].

## Results

**Structure of Intasome-NCP strand-transfer complex**. To understand intasome strand transfer into NCPs, we assembled the

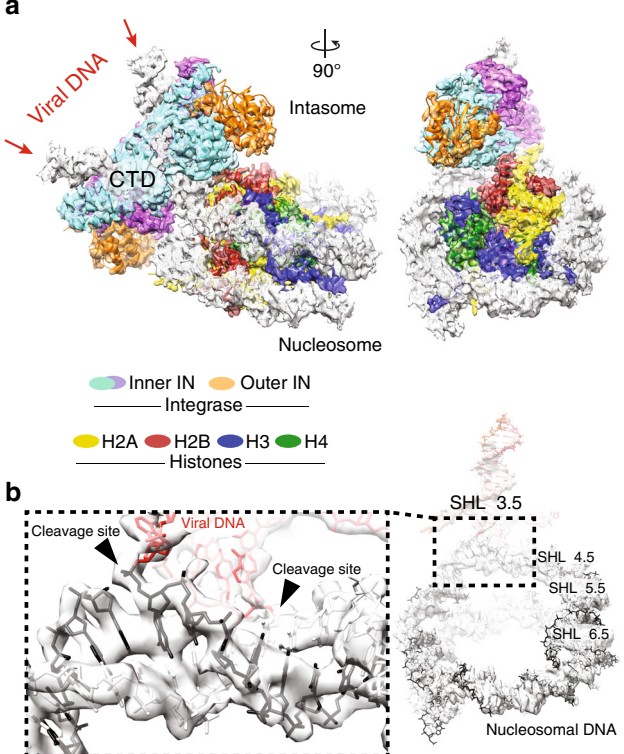

**Fig. 1** Intasome-NCP strand-transfer complex visualised by cryo-EM. **a** 3.9-Å resolution structure of the post-catalytic intasome–NCP complex. IN means intasome, CTD means C-terminal domain. **b** Covalently linked viral (red) and nucleosomal (black) DNA. Integration occurred at the superhelical location (SHL) location 3.5

complex of the PFV intasome and the NCP containing a native human DNA sequence (termed D02), previously selected for its ability to form a stable PFV–NCP complex[15]. Following isolation by size exclusion chromatography, the intasome-NCP complex was incubated in the presence of $Mg^{2+}$ to facilitate strand transfer[15]. We then used cryo-EM imaging and single-particle approaches to determine the structure of the resulting post-catalytic assembly to 3.9 Å resolution (Supplementary Fig. 1, Table 1). Docking known crystallographic coordinates into the cryo-EM map, manual adjustment, and real-space refinement allowed us to generate an atomic model of the Intasome-NCP strand transfer complex.

As previously observed, intasome engages the strongly preferred site on the nucleosomal DNA, at SHL 3.5[15,24] (Fig. 1). The new structure is overall similar to the original lower-resolution intasome-NCP complex, which was captured in the pre-catalytic state (Fig. 1a), confirming that strand transfer is not accompanied by large conformational rearrangements[6]. According to the atomic model, at the integration site, DNA is lifted by 7 Å from the histone octamer and bent to allow access to the IN catalytic centre, in excellent agreement with the earlier observations based on the crystal structures of the PFV strand transfer complex[6,16,25] and the lower-resolution intasome-NCP cryo-EM data[15]. Local resolution ranges between ~3.5 Å throughout the histone octamer core, and ~4–4.5 Å for nucleosomal DNA, similar to other NCP structures determined by cryo-EM (Supplementary Fig. 1)[26–28]. Nevertheless, we could confidently model the DNA phosphate backbone for the entire assembly. The integration site on the nucleosomal DNA is sandwiched between the histones and the intasome, resulting in a higher local resolution ( ~3.7 Å). Notably, a discontinuity in the cryo-EM

**Table 1 Data collection and processing information**

| Parameter | Intasome-NCP | NCP-D02-strep | 601 nucleosome |
|---|---|---|---|
| **Data Collection** | | | |
| Microscope | FEI Titan Krios | FEI Titan Krios | FEI Titan Krios |
| Detector | FEI Falcon II | FEI Falcon III | FEI Falcon III |
| Acceleration voltage (kV) | 300 | 300 | 300 |
| Number of micrographs | 4916 | 4182 | 1300 |
| Frames per micrographs | 7 | 30 | 30 |
| Frame rate (/s) | 4.3 | 60 | 60 |
| Dose per frame (e-/pixel) | 9.86 | 1.12 | 1.24 |
| Accumulated dose (e-/Å$^2$) | 56 | 28.3 | 31.3 |
| defocus range (μm) | 1.5–3.5 | 1.5–3.5 | 1.5–3.5 |
| **Frames** | | | |
| Alignment software | MotionCorr | MotionCor2 | MotionCor2 |
| Frames used in final reconstruction | 1–7 | 1–30 | 2–30 |
| Dose weighting | No | yes | yes |
| **CTF** | | | |
| Fitting software | CTFFIND3 | Gctf | Gctf |
| Correction | full | full | full |
| **Particles** | | | |
| Picking software | Xmipp & Relion 1.3 | Relion 2.1 | Relion 2.1 |
| Picked | 989177 | 1131653 | 205680 |
| Used in final reconstruction | 177155 | 62196 | 123123 |
| **Alignment** | | | |
| Alignment software | Relion 1.3 | Relion 2.1 | Relion 2.1 |
| Initial reference map | EMD-2992 | CryoSPARC *ab initio* | CryoSPARC *ab initio* |
| low pass filter limit (Å) | 50 | 50 | 50 |
| number of iterations | 25 | 25 | 25 |
| local frame drift correction | yes | no | no |
| **Reconstruction** | | | |
| Reconstruction software | Relion 1.3 | Relion 2.1 | Relion 2.1 |
| Box Size | 240 × 240 × 240 | 256 × 256 × 256 | 256 × 256 × 256 |
| Voxel size (Å) | 1.11 | 1.09 | 1.09 |
| Symmetry | C1 | C1 | C2 |
| Resolution limit (Å) | 2.22 | 2.18 Å | 2.18 Å |
| Resolution estimate (Å) | 3.9 | 4.2 | 3.5 |
| Masking | Yes | Yes | Yes |
| Sharpening (Å$^2$) | Bfactor: -146 | Bfactor: -150 | Bfactor: -110 |
| EMDB ID | EMD-4960 | EMD-4692 | EMD-4693 |
| **Model building** | | | |
| Number of protein residues | 1742 | 747 | |
| Number of DNA residues | 358 | 284 | |
| Bond length outliers | 0.00% | 0.00% | |
| Bond angle outliers | 0.02% | 0.00% | |
| Bonds (R.M.S.D) | 0.010 | 0.008 | |
| Angles (R.M.S.D) | 1.183 | 0.856 | |
| Ramachandran favoured/outlier | 94.3%/0.00% | 96.85%/0% | |
| Rotamer favoured/outlier | 98.5%/0% | 99.51%/0% | |
| Clashscore | 10.55 | 4.91 | |
| Model vs Data CC (mask) | 0.71 | 0.85 | |
| Molprobity score | 1.91 | 1.45 | |
| PDB ID | PDB: 6RNY | PDB: 6R0C | |

density resulting from the nucleosomal DNA cleavage at the site of integration (Fig. 1b) confirms that strand transfer has indeed occurred in our nucleoprotein assembly as observed biochemically[15] (Supplementary Fig. 2).

Intasome engages nucleosomal DNA non-symmetrically at two distinct sites: at the strand transfer site, as well as at the opposing gyre, which nestles in the cleft between one catalytic and one outer IN subunit (Fig. 1a). Near the integration site, the C-terminal alpha-helix of histone H2B makes direct contact with the C-terminal domain of one catalytically competent IN subunit, providing corroborating evidence for the previously reported role of PFV IN residues Pro135, Pro239 and Thr240 in engaging the C-terminus of H2B[15]. Furthermore, the higher quality of the new cryo-EM map allowed us to build a backbone model for a segment of the N-terminal H2A tail, revealing intimate contacts

of H2A Lys-9 and Arg-11 with the IN C-terminal domain (Fig. 2a). Concordantly, truncation of the first 12, but not 8 H2A residues lead to a reduction of intasome-NCP complex formation (Fig. 2b). Furthermore, Ala substitutions of either H2A at Lys-9 or Arg-11 affect complex stability, while a combination of the two substitutions fully abrogated stable complex formation under conditions of the pull-down assay (Fig. 2c).

**Asymmetric reconstruction of the human D02 NCP**. Similar to the pre-catalytic complex, our new structure of an intasome-NCP strand-transfer complex features a nucleosomal DNA loop bulging away from the protein octamer by ~7 Å at the integration site. Although occurring at a different superhelical location, the DNA looping is reminiscent of structures of NCPs engaged by

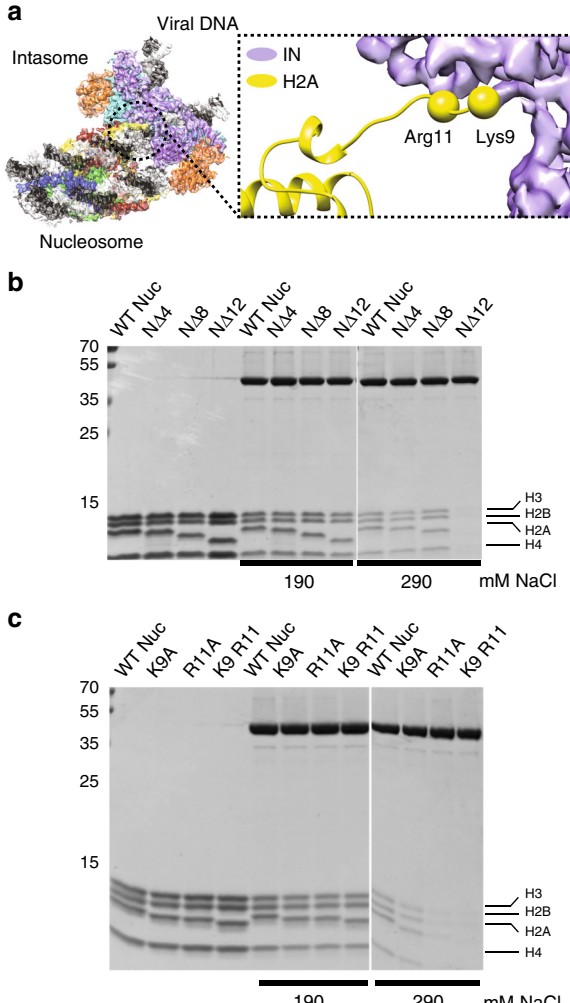

**Fig. 2** Intasome interaction with the N-terminal histone H2A tail. **a** Histone H2A residues Lys-9 and Arg-11 play a role in the intasome-NCP interaction. **b**, **c** Pull-down with immobilised intasome binding to NCP bearing deletions **b** or amino acid substitutions **c** in the H2A N-terminal tail. NΔ4, NΔ8, NΔ12, NCP variants lacking the first 4, 8 or 12 residues fo histone H2A

chromatin remodelers such as SWR1. Interestingly, DNA looping by SWR1 is accompanied by both sliding of nucleosomal DNA, as well as histone octamer distortion[22]. We wanted to test whether intasome-induced looping is compensated by nucleosomal DNA sliding along the histone octamer, as observed for chromatin remodelers. To this end, we decided to directly compare the cryo-EM structure of the intasome-NCP strand-transfer complex with that of an isolated NCP, containing the same native human D02 nucleosomal DNA sequence[15].

Reconstructing a D02 NCP presented a number of significant challenges. Firstly, the NCP containing D02 DNA is less stable than NCPs wrapped with strongly positioning sequences such as Widom 601[15,29]. Our EM analysis of the isolated NCP D02 revealed that, unlike the intasome complex, D02 NCPs had the tendency to become unravelled, especially in the presence of higher salt measured by the lack of NCP particles in holey grids. However, exposure to mild crosslinking conditions (0.05% glutaraldehyde, 5 min, 4 °C) yielded tractable particles that were visible on open-hole cryo grids. Importantly, mild NCP-crosslinking did not prevent intasome activity as measured in strand-transfer assays (Supplementary Fig. 2). A second challenge was presented by the asymmetry of the D02 DNA sequence,

which leads to the strongly preferred intasome capture at one side of the NCP[15]. Thus, to describe any intasome-dependent sliding along the histone octamer, we first had to reconstruct the D02 NCP avoiding two-fold averaging. However, both the histone octamer and the DNA backbone contain a prominent two-fold symmetric character, which strongly influence particle alignment and prevent asymmetric reconstruction. To facilitate asymmetric particle alignment, we introduced a biotin moiety on the end of the DNA arm distal from the integration site and decorated NCPs with streptavidin (Fig. 3a). Critically, streptavidin attachment did not affect NCP stability, nor the ability of intasome to integrate into NCPs (Supplementary Fig. 2). Crosslinked D02 NCPs, imaged by cryo-EM and analysed by two-dimensional (2D) averaging, revealed multiple views of the coin-shaped NCP assemblies (Fig. 3b). Particles appeared decorated by diffuse density projecting from one DNA arm, which we assigned to streptavidin. Free streptavidin particles ( ~ 75 kDa) could also be identified amongst the 2D class averages (Supplementary Fig. 3). Next, we used single-particle reconstruction to determine the 4.2-Å resolution structure of NCP-D02-streptavidin complex (Supplementary Fig. 3 and 4). As the streptavidin is linked to the 5′ end of a distal DNA arm, it is less ordered than the rest of the assembly, and appears not to be engaged in interactions with the NCP core (Fig. 3c, Supplementary Fig. 3). Therefore, streptavidin helps align particles asymmetrically while seemingly not interfering with the NCP structure.

Originally selected from a genome-wide screen for strong intasome interactors, the D02 DNA sequence allowed isolation of a mono-disperse intasome–NCP complex[15]. Detailed inspection of the isolated D02 NCP cryo-EM maps provides insight into intasome selectivity. Firstly, nucleosomal DNA arms appear to be flexible (as detected by inspection of the local resolution map reported in Supplementary Fig. 3, and given the significant number of unwrapped NCPs averaged during analysis). We asked whether the same flexibility could be observed for a NCP containing a strong positioning sequence such as Widom 601, which while serving a good target for strand transfer did not allow formation of long-lived pre-catalytic intasome–NCP complex[15]. To this end, we solved a 3.5-Å resolution cryo-EM structure of a Widom 601-wrapped nucleosome containing strongly positioned Widom 601 sequence with 13-bp long linker DNA arms (Supplementary Fig. 5). Only linker DNA fragments display a degree of flexibility in the Widom 601 structure. We postulated at this stage that the flexibility of the D02 NCP DNA arms might facilitate DNA looping, prompting us to further investigate the mechanism.

Another notable feature of the D02 NCP is the limited interaction between DNA and the N-terminal tail of H2A, reflected by poorly defined density contacting nucleosomal DNA at SHL 4.5. This differs for example from our structure of Widom 601 NCP, which shows discrete ordering of H2A N-terminal tail in the minor groove of nucleosomal DNA at the equivalent position, in agreement with previous crystallography and cryo-EM studies[7,30–32]. We speculate that a loose DNA engagement renders the histone H2A tail available for intasome binding as observed in our strand-transfer complex, hence improving substrate selection (Fig. 3d).

**Retroviral integration shifts nucleosomal DNA register.** To understand the impact of retroviral integration on NCP architecture, we analysed the structural changes in the NCP that accompany productive engagement with the intasome. Comparison of the intasome-D02 NCP structures prior to and after strand transfer shows that histones undergo relatively minor distortions clustered around the histone H3-H4 dimer on the nucleosomal face proximal to the integration site (Supplementary

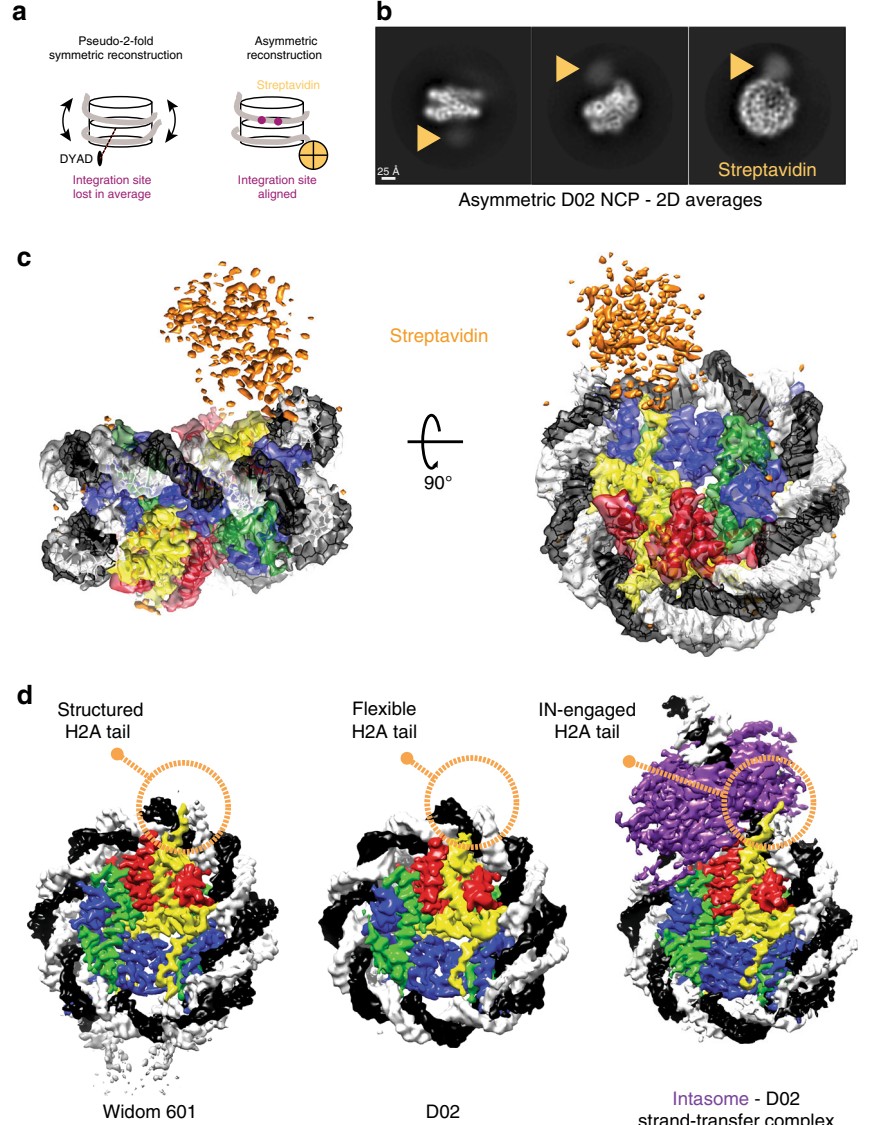

**Fig. 3** Asymmetric reconstruction of the isolated D02 NCP. **a** Streptavidin labelling allows asymmetric reconstruction of the D02 NCP, which avoids pseudo-two-fold symmetry averaging of the integration-site DNA. **b** 2D averages of labelled D02 NCP particles reveal a discernible, diffuse density for streptavidin at the nucleosomal DNA arm distal from the integration site. **c** 3D reconstruction of D02 NCP reveals an asymmetric streptavidin label decorating one of two nucleosomal DNA arms. **d** Unlike for Widom 601, the D02 NCP contains limited density for the N-terminal H2A tail, indicating that this element is flexible and available for intasome engagement. In fact, in the Intasome-NCP strand-transfer complex the N-terminal H2A tail interacts with IN, stabilising the interaction (also see Fig. 2b, c)

Fig. 6). Conversely, in our atomic model DNA looping at the integration site is compensated by a significant change in nucleosomal DNA register, with the nucleosomal DNA arm proximal to the integration site shifting by 2 bp (Fig. 4a). This shift in register extends from SHL 7 to SHL 2.5, where an interaction with H3 element L1 appears to hold DNA in place and limit downstream sliding of the double helix (Fig. 4b, Supplementary Movie 1).

To validate the DNA-register change observed in our structural models, we turned to a single-molecule FRET assay. We used a Cy3 donor to label the 5′-terminal end of the nucleosomal DNA closest to the integration site, and a Cy5-maleimide-cysteine acceptor engineered at position 119 of H2A (Fig. 5a). Histone labelling was optimised to yield approximately one fluorophore per octamer. Surface-immobilised NCPs were imaged by FRET in the absence or presence of the intasome and/or $Mg^{2+}$ (Fig. 5b). In reconstituted NCPs, single H2A labels were found either

proximal to, or distal from, the Cy3-modified DNA end. The main energy transfer group deriving from the proximal fluorophore pair centred around 0.95 FRET efficiency, while the second distal fluorophore pair peak centred around 0.37 transfer efficiency (Supplementary Fig. 7A). We focused our analysis on the 0.95 FRET group, as any shift in nucleosomal DNA register would cause more pronounced changes in FRET efficiency in this population. In all tested conditions, FRET efficiency was stable, with a minor population (~10%) of traces exhibiting slight changes in FRET intensity (Fig. 5c, Supplementary Fig. 7B). Supplementing the NCP with intasome or $Mg^{2+}$ did not result in any significant FRET change (Fig. 5d, e). However, when strand transfer was induced by adding both intasome and $Mg^{2+}$, a separate, ~0.8 FRET population appeared (Fig. 5f). This second population is consistent with a shift in register of the DNA moving away from the K119C-Cy5 H2A residue (Fig. 5a). These results are in good agreement with our comparative cryo-EM

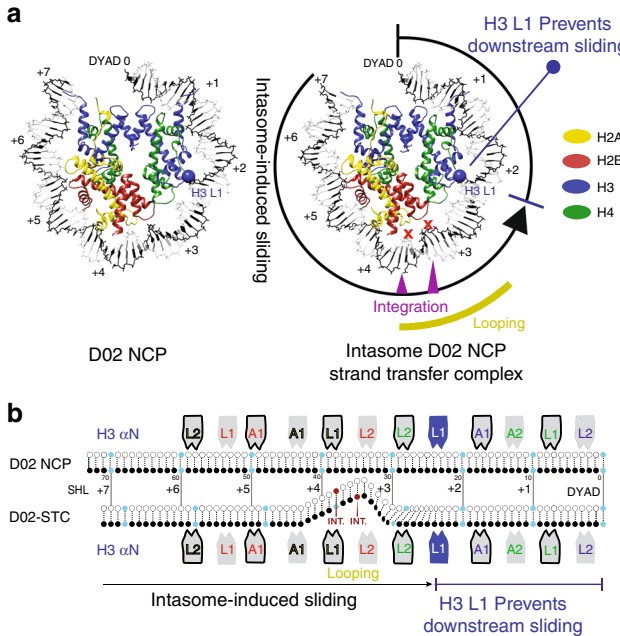

**Fig. 4** Integration-promoted DNA sliding observed by cryo-EM. **a** On the left, slabbed view of the isolated D02 NCP. Histone H3 L1 element is highlighted with a blue ball. On the right, DNA looping required for retroviral integration causes a shift in the DNA register, which extends from SHL 7 to 2. Histone H3 L1 element prevents downstream DNA sliding. **b** Schematic representation of integration-induced NCP remodelling

analyses indicating that intasome-mediated looping required for integration promotes sliding of nucleosomal DNA (Fig. 4a). In fact, the observed drop in FRET efficiency indicates a small but significant shift in the DNA register that corresponds to less than 4 bp, according to a calibration previously obtained with Widom 601 NCPs[22]. Crystallographic and cryo-EM structures of pre-catalytic assemblies of intasome bound to DNA or nucleosomes established that target capture alone leads to DNA bending and nucleosomal DNA remodelling[15,16]. The new post-catalytic intasome-NCP structure reported shows no change of DNA looping at the integration site after strand transfer. However, in our single molecule experiments, a drop in FRET efficiency was only observed in the presence of $Mg^{2+}$ required for catalysis. Although this observation was initially surprising to us, we note that the precatalytic intasome-nucleosome complex used in our earlier work[15] was purified under elevated ionic strength conditions to enrich for higher affinity productive interaction[33]. Indeed, of the two symmetry-related D02 SHL ±3.5 intasome binding sites, near equally targeted in a bulk strand transfer assay, only one was occupied in the purified material[15]. Conditions of the single molecule FRET used here were more similar to the bulk strand transfer assay, allowing for detection of transient interactions and fixation of productive complexes in the absence and presence of $Mg^{2+}$, respectively. We speculate that most intasome complexes observed in the absence of the metal ion cofactor result from transient scanning[15] interaction, which alone is unlikely to result in DNA deformation. Alternatively, DNA perturbation and sliding could be functionally distinct steps. This could be mediated by histone buffering of DNA displacement, as observed previously[34]. In this model, tension in the lifted DNA at the integration site could be partially accommodated by protein–DNA contacts within the target capture complex, without a immediate shift in DNA, which would occur upon full strand transfer catalysed upon addition of $Mg^{2+}$.

## Discussion

Over the last 35 years macromolecular crystallography has provided several high-resolution views of the NCP and its binding partners. These efforts led to describing the NCP architecture at an atomic level[7–9], explained how DNA sequence can influence wrapping of the double helix[35], and how common docking sites on the histone octamer are recognised by different interactors[36–40]. Over the last four years, cryo-EM has started to provide a dynamic view of the NCP[26,41–44]. Recent data indicated that NCPs are more flexible in solution, with the histone octamer visiting more compacted or extended states, compared with a nucleosome trapped in a crystal lattice[45]. NCP unwrapping has been visualised with cryo-EM, for example in the context of the hexasome, which is an NCP with partially unpeeled DNA, due to the loss of one H2A/H2B dimer[28]. Spectacular views of progressively unwrapped NCPs have been obtained for transcribing RNA polymerase II captured during NCP passage[46,47]. Moreover, cryo-EM provided the first glimpses of ATP-dependent NCP translocation through a mechanism involving DNA looping and sliding along the histone octamer[22,41,48–54].

Our high-resolution view of a post-catalytic intasome–NCP complex provides an example of a local remodelling of nucleosomal DNA. Although previous work established formation of a DNA loop during productive intasome–NCP interaction, it was not clear whether the loop is accommodated by partial underwinding of flanking DNA or through a shift in nucleosomal DNA register. Because IN must catalyse only one strand transfer event and does not need to cycle between states on the chromatin, it does not depend on a power source, unlike ATP-driven translocases and nucleosome remodelers[34]. Therefore, all conformational DNA rearrangements are offset by energy released with the formation of the intasome-NCP interface. Nevertheless, similarities with the mechanism of DNA translocation of chromatin remodelers can be identified. In fact, in both systems, DNA is looped out of the histone core, causing a compensatory register shift of the double helix wrapped around the octamer. Nucleosomal DNA looping at SHL 3.5 is required for access to the IN active site[15], and causes DNA sliding around the histone octamer, with global repositioning extending from SHL 7 to SHL 2. At this site, histone H3 element L1 holds the sugar-phosphate backbone in place, preventing any further downstream shift in DNA register (Fig. 4b, Supplementary Movie 1). Using cryo-EM, Kurumizaka and colleagues have recently shown that the same H3 L1-DNA interaction stalls RNA polymerase II during nucleosome passage[46]. ATP-powered translocases such as Swr1 and Snf2 have been observed to engage and loop out SHL 2 DNA, disrupting the H3 L1-DNA interaction[22,48,55]. It is tempting to speculate that the concerted action of intasome and SHL 2 remodelers could act synergistically during DNA unpeeling and strand-transfer complex disassembly, required to complete retroviral integration.

## Methods

**Intasome purification.** The intasome was assembled using recombinant PFV IN and double stranded synthetic oligonucleotides mimicking the pre-processed U5 end of the vDNA as previously described[4,15]. Briefly, hexahistidine-tagged IN was overexpressed in BL-21 CodonPlus RIL cells (Agilent). Cells were lysed in 25 mM Tris–HCl pH 7.4, 0.5 M NaCl, 1 mM PMSF by sonication; clarified lysate supplemented with 20 mM imidazole was applied to packed, equilibrated Ni-NTA resin (Qiagen). The resin and washed extensively in lysis buffer supplemented with 20 mM imidazole. Bound proteins were eluted with lysis buffer supplemented with 200 mM imidazole and protein-containing fractions were supplemented with 5 mM DTT. The hexahistidine-tag was cleaved by incubation with human rhinovirus 14 3 C protease. The protein, diluted to reduce the NaCl concentration to 200 mM, was loaded onto a HiTrap Heparin column (GE Healthcare). IN was eluted using a linear gradient of 0.25-1 M NaCl. IN-containing fractions were concentrated and further purified by size exclusion chromatography through a Superdex-200 column (GE Healthcare), equilibrated in 25 mM Tris pH 7.4, 0.5 M NaCl. Protein, supplemented with 10% glycerol and 10 mM DTT, was

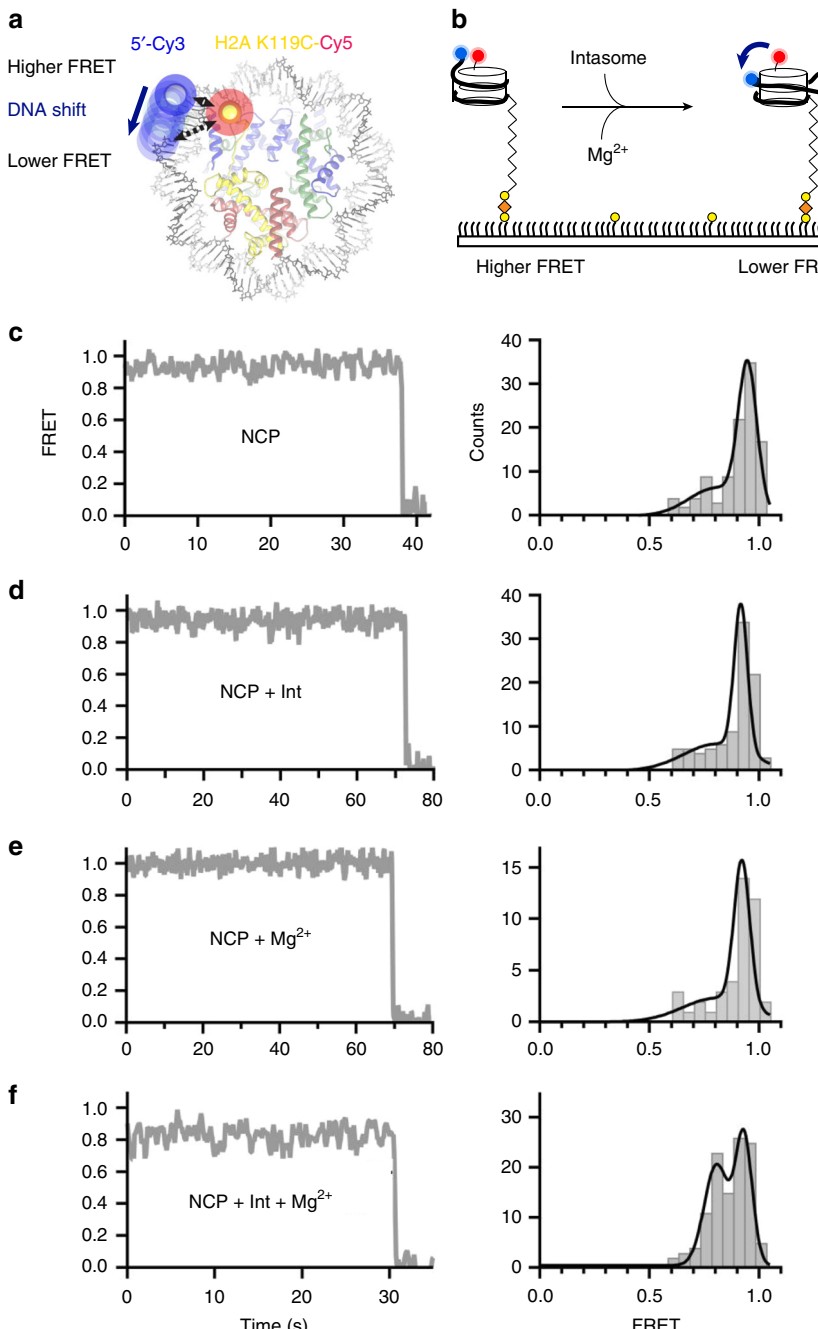

**Fig. 5** Integration-promoted nucleosomal DNA sliding observed by single molecule FRET. **a** Fluorescently labelled NCP: labelled octamer (**Cy5** on H2A, red) wrapped with biotinylated labelled DNA (**Cy3** on exit site, blue). **b** NCPs are surface-immobilised on neutravidin (orange) coated biotin-PEG (yellow) slides. Intasome-induced translocation in the presence of magnesium is detected as a FRET decrease. Representative single-molecule FRET trajectories (left) and histograms (right) of proximal-only labelled NCPs **c** without intasome or magnesium ($N = 105$), **d** in the presence of intasome ($N = 93$), **e** in the presence of magnesium ($N = 42$), and **f** in presence of both intasome and magnesium ($N = 115$). A significant population shift to ~ 0.8 FRET in the presence of both intasome and magnesium. Data were collected at 100 ms/frame (10 Hz) and smoothed with a 3-point moving average. Black lines in FRET histogram are fits to two Gaussian distributions

concentrated to 10 mg/ml, as estimated by spectrophotometry at 280 nm and stored at −80 °C.

To assemble the intasome a mixture containing 120 μM PFV IN and 20 μM pre-annealed DNA oligonucleotides 5′-TGCGAAATTCCATGACA and 5′-ATTGTCATGGAATTTCGCA (IDT) in 500 mM NaCl was dialysed against 50 mM BisTris propane-HCl pH 7.45, 200 mM NaCl, 40 μM ZnCl$_2$, 2 mM DTT for 16 h at 18 °C. A list of all oligos is provided in Supplementary Table 1. Following dialysis, the assembly reaction, supplemented with NaCl to a final concentration of 320 mM, was incubated on ice for 1 h prior to purification on Superdex-200 column in 25 mM Bis-Tris propane-HCl pH 7.45, 320 mM NaCl. Purified intasome, concentrated by ultrafiltration, was kept on ice for immediate use.

**NCP formation**. NCPs were assembled essentially as described[15,56]. Briefly Human H2A, H2A K119C, H2B, H3.3, H3.1 C96SC110A and H4 were over-expressed in *E. coli* and purified from inclusion bodies. Histones were refolded from denaturing buffer through dialysis against 10 mM Tris-HCl pH 7.5, 2 M NaCl, 5 mM beta-mercaptoethanol, 1 mM EDTA buffer, and octamers were purified by size exclusion chromatography over a Superdex-200 column (GE Healthcare). DNA fragments for wrapping NCPs (171-bp Widom 601 DNA, 145-bp D02 DNA or D02 DNA appended with biotin and fluorophores) were generated by PCR using Pfu polymerase and HPLC-grade oligonucleotides (IDT). PCR products generated in 96-well plates (384 × 100 μl) were pooled, filtered and purified on a ResorceQ column as described[15]. NCPs were assembled by salt dialysis as described[15,30,56] and heat

repositioned at 37 °C for 30 min. D02 containing NCPs were further purified using a PrepCell apparatus with a 5% polyacrylamide gel (BioRad).

**NCP-streptavidin complex formation**. Purified *Streptomyces avindii* streptavidin powder (Sigma-Aldrich) was resuspended in 20 mM HEPES-NaOH pH 7.5, 150 mM NaCl at a final concentration of 35 μM. A derivative of D02 DNA was used for NCP reconstitution, containing a 5′ biotin moiety on the exit arm distal from the intasome-engagement site. To form the NCP-streptavidin complex, biotinylated D02 NCP (0.5 μM) was incubated with 0.3 μM streptavidin for 10 minutes at room temperature in 20 mM HEPES pH7.5, 150 mM NaCl, 1 mM DTT, 1 mM EDTA.

**EM sample preparation**. The intasome-DO2 NCP complex was formed and purified by size exclusion chromatography as previously described[15]. Briefly, 200 μg of D02 NCP and 200ug of PFV intasome were mixed in 25 mM Bis-Tris Propane, 320 mM NaCl, prior to application on Superdex 200 10/300 column. To allow strand transfer, the complex was incubated in the presence of 5 mM MgCl$_2$ for 30 min at room temperature. Cryo-EM sample preparation was performed as follows: 4 μl of the integration reaction were applied to plasma cleaned C-Flat 1/1 400 mesh grids; after 1 min incubation, grids were double side blotted for 3.5 s using a CP3 cryo-plunger (Gatan), operated at 80% humidity, and quickly plunge-frozen into liquid ethane. Ice quality was checked using a JEOL-2100 Lab6 operated at 120 kV, using a 914 side-entry cryo-holder (Gatan), and images were recorded on an UltraScan 4kx4k camera (Gatan). The best cryo-grids were retrieved, stored in liquid nitrogen and later shipped in a dry-shipper to NeCEN (University of Leiden, The Netherlands). At NeCEN, grids were loaded into a Cs corrected Titan Krios microscope and the data was collected over two different sessions using the EPU software (ThermoFisher Scientific). Images were recorded at a nominal magnification of 59,000 X on Falcon II direct electron detector yielding a pixel size of 1.12 Å /pixel with a defocus range of −1.5 to −3.5 μm. Data were collected as movies of 7 frames over 1.6 s giving a total applied dose of 56 electrons/Å$^2$. A total of 4,916 movies were collected.

The D02 biotin-streptavidin complex was gently cross-linked with 0.05% glutaldehyde at room temperature for 5 min, prior to quenching with 50 mM TrisHCl pH 7.5. The complex was concentrated and buffer exchanged using a 50-kDa MWCO spin concentrator (Amicon) into 10 mM Tris-HCl pH 7, 20 mM NaCl, 1 mM EDTA, 1 mM DTT; 3.5 μl sample at 80 ng/μl (DNA concentration based on spectrophotometry) was applied to Quantifoil 2/2 grids, with fresh carbon pre-evaporated onto the grids to better control ice thickness. Grids were glow discharged at 40 mA for 1 min. Sample was blotted in a Vitrobot Mark IV using -1 offset, 15 s wait time and 2.5 s blot at 4 °C and 100% humidity, before plunge-freezing in liquid ethane. Grids were stored in liquid nitrogen prior to loading on a Titan Krios operated at 300 kV. Data was acquired using a Falcon III detector operating in counting mode using a pixel size of 1.09 Å, a total dose of 30 electrons/Å$^2$ and a defocus range from -1.5 to -3.5 μm. A total of 4,182 movies were collected automatically using the EPU software (ThermoFisher Scientific). The Widom 601 NCP sample was applied to freshly glow discharged Quantifoil 2/2 grids and sample was blotted in a Vitrobot Mark IV using -1 offset, 10 s wait time, 3.5 s blot at 4 °C and 100% humidity, before plunge-freezing in liquid ethane. Data were acquired using a Falcon III detector operating in counting mode using a pixel size of 1.09 Å and total dose of 30 electrons/Å$^2$. A total of 1,300 Micrographs were collecting using automated EPU software.

**Cryo-EM image processing**. For the intasome-DO2 NCP complex dataset (Supplementary Fig. 1), movie frames were corrected for to beam-induced drift[57] and a sum of each aligned movie was used in the first steps of image processing. All movies showing any remaining drift or containing ice were discarded at this stage, and only the best 3,125 movies were selected for further image processing. First, 989,177 particles were automatically picked using Xmipp[58] and Relion version 1.3[59]. Contrast transfer function parameters were estimated using CTFFIND3[60], and all 2D and 3D classifications and 3D refinements were performed using RELION[59]. After 2 rounds of 25 iterations of 2D classification, 335,989 particles remained and were subjected to 3D classification using the pre-catalytic intasome-NCP map[15], filtered to 50 Å resolution, as a starting model. To speed up calculations, 8 classes were generated with a 15° angular sampling. The best 3 classes were merged into one 232,000 particles dataset. 3D refinement of this subset yielded a 4.7 Å map. A second round of 3D classification step was performed with 4 classes and a finer 7.5° angular sampling. The best 3 classes were merged together for a total of 177,155 particles. Refinement of this dataset yielded a 4.2 Å map. Statistical movie processing was then performed, as described previously[61] and the resulting map reached 3.9 Å resolution after correction for the modulation transfer function and sharpening[62]. Resolutions are reported according to the "gold-standard" Fourier Shell Correlation, using the 0.143 criterion[63].

For the D02-NCP-Streptavidin and Widom 601 NCP datasets (Supplementary Figs. 3-5) all micrographs were motion-corrected using MotionCorr2 using all frames (D02-NCP-Streptavidin) or removing the first frame (Widom 601 NCP). CTF parameters were estimated using Gctf[64] and poor micrographs were discarded. Particles were picked in RELION-2.1 using reference classes obtained from a manually-picked, 50-micrograph dataset. Two rounds of 2D classification

were performed to discard poorly averaging particles. 3D classification was performed using a 50 Å, low pass filtered initial model, based on results from an *ab initio* reconstruction derived from cryoSPARC[65]. For the Widom 601 NCP, particles contributing to 3D classes with discernible secondary-structure features were pooled and refined using a spherical mask, and postprocessed in RELION-2.1[66] resulting in a 3.8 Å (C1 symmetry applied) or 3.5 Å resolution (C2 symmetry applied). For the D02-NCP-Streptavidin, a relatively smaller percentage of particles contributed to subnanometre-resolution 3D averages. This is likely because of evident flexibility of the both the exit nucleosomal DNA and the streptavidin group. To help drive streptavidin alignment and avoid artificial NCP symmetrisation, a loose mask was used in a subsequent round of 3D classification, encompassing both NCP and streptavidin. The resulting asymmetric reconstruction yielded a reconstruction with 4.6 Å (no mask) or 4.2 Å resolution (loose mask applied during refinement).

**Atomic model docking and refinement**. For the NCP-intasome STC complex NCP (3UTB[67] from PDBredo) and PFV strand transfer complex (3OS0[16]) crystal structures were docked in the EM map using Chimera[68] and clashing DNA segments were removed from the model. In order to refine the voxel/pixel size of the map a series of maps were calculated with voxel/pixel size from 0.9 to 1.15 in steps of 0.01 and the initial model was refined against each map using phenix.real_space_refine[69] with no additional geometry restraints. The geometry of resulting models was compared, and voxel/pixel size fine-tuned between 1.11 and 1.12 in steps of 0.001. The model refined against the map with voxel/pixel size of 1.111 maintained the best geometry and was used for further model building and refinement. The model was adjusted, and sequence of protein and DNA components matched to the biological sample manually in Coot[70] and refined using phenix.real_space_refine (Nightly build version 1.10pre-2091)[71] and Namdinator[72,73]. Additional restraints describing protein secondary structure, DNA base pairing and stacking were used in Phenix. Protein geometry was assessed with Molprobity[69] and DNA geometry was assessed with 3DNA[74]. For the D02 structure, NCP structure 5MLU was used as the starting model to be independent from the NCP-intasome STC structure. The sequence was adjusted and model manually tweaked in Coot and refined using phenix.real_space_refine (Nightly build version phenix-dev-3374). Fine tuning of the voxel/pixel size was deemed unnecessary as the model refined without issue. Both models have reasonable stereochemistry and are in good agreement with the EM maps.

**Single-Molecule FRET experiments**. Doubly-labelled nucleosomes were generated with a biotin on distal exit DNA and a single fluorophore donor (Cy3) attached on the proximal exit DNA end, and the acceptor fluorophore (Cy5) at H2A position 119. To generate protein-Cy5-labelled octamers H2A K119C was incorporated into octamers with H3.1 C96SC110A, H2B and H4 as described above, with an additional desalting step in a Zeba Spin column (ThermoFisher, 7 K MWCO) to remove beta-mercaptoethanol. Octamers at 70 μM (140 μM of cysteine) were incubated with 5 mM TCEP for 10 min at room temperature. To achieve partial labelling, sulpho-Cy5 maleimide was added at 105 μM for 1 hour at room temperature. The reaction was quenched by adding 5 mM beta-mercaptoethanol and desalted to remove unreacted dye (ThermoFisher, 7 K MWCO). The extent of labelling was quantified by measuring the 595 nm/280 nm absorbance ratio, as well as by 2D intact mass ESI mass spectrometry, with an estimated labelling efficiency of 68%. D02 DNA was generated by PCR, using oligos containing Biotin-TEG-C18 and Cy3 modifications attached to the 5′ termini. The PCR product was purified as described above. Nucleosomes were reconstituted as described above.

Single-Molecule FRET experiments were performed with freshly purified intasome complex. Quartz slides and coverslips were cleaned and passivated with methoxy-PEG-SVA (M$_r$ = 5,000, Laysan Bio, Inc.) containing 10% biotin-PEG-SVA (M$_r$ = 5,000, Laysan Bio, Inc.) in 100 mM sodium bicarbonate, and used to construct a microfluidic channel as described previously[75]. Neutravidin (0.2 mg/ml in 50 mM Tris-HCl, pH 7.5, and 50 mM NaCl) was injected in and incubated for 5 min. Excess neutravidin was washed out with intasome buffer (25 mM bis-Tris propane, pH 7.45, 240 mM NaCl, 4 μM ZnCl$_2$ and 1 mM DTT). Biotinylated fluorescently labelled nucleosomes in intasome buffer containing 0.2 mg/ml BSA were surface immobilised by incubation in the microfluidic channel for 5 min. Excess nucleosomes were washed out and immobilised nucleosomes imaged in imaging buffer composed of intasome buffer in addition to 2 mM Trolox, oxygen scavenging system (2.5 mM 3,4-dihydroxybenzoic acid, 250 nM protocatechuate dioxygenase) and 0.2 mg/ml BSA. Experiments were performed in the absence and presence of 500 nM intasome and 5 mM magnesium. Fluorescent molecules were imaged using a custom-built prism-based total-internal reflection fluorescence (TIRF) microscope[76]. All measurements were recorded at room temperature (21ºC) using continuous green laser (532 nm, 2.5 mW) excitation at 100 ms time resolution. Apparent FRET efficiencies were calculated as the ratio of acceptor intensity divided by the sum of acceptor and donor intensities. FRET histograms of labelled nucleosomes were obtained by calculating the mean FRET efficiency of 40–100 trajectories from multiple fields of view, as stated in the figure captions. All experiments have been performed at least twice, on different days and with different combinations of protein preps.

**Intasome strand-transfer and pull-down assays**. Intasome integration assays were performed as described[15], briefly 5 µg of NCPs were incubated with 1.5 µg of intasome in intasome reaction buffer with and without 5 mM MgCl$_2$ at 37 °C for 15 min. The reaction was quenched by the addition of 25 mM EDTA and 0.2% SDS, and DNA precipitated after proteinase K digestion. DNA was then separated on 4–12% TBE polyacrylamide gels. Pull-down assays were performed, as previously described[15]. Briefly 10 µg of biotinylated intasome was incubated with 10 µg of NCP variants in pull-down buffer with increasing concentrations of sodium chloride. PFV and associated NCPs was immobilised on streptavidin beads (Life technologies), washed extensively and eluted by heating at 37 °C in 1.3× SDS Laemmeli buffer. All experiments have been performed at least twice, on different days and with different combinations of protein preps.

**Reporting Summary**. Further information on research design is available in the Nature Research Reporting Summary linked to this article.

## Data availability

Model coordinates for the NCP-D02-streptavidin and Intasome-NCP structures are deposited in the Protein Data Bank under accession code 6RNY and 6R0C respectively. Cryo-EM maps for NCP-D02-streptavidin, NCP-601 and Intasome-NCP are available at the EMDB under codes EMD-4692, EMD-4693 and EMDB-4960 respectively. The source data underlying Fig. 2b and c, 5c–f and 7c and Supplementary Figs 2, 3a, 5a, 6a and 7 are provided as a Source Data file. Other data are available from the corresponding authors upon reasonable request.

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

## Acknowledgements

We thank Rishi Matadeen (formerly at NeCEN) for data collection of the NCP-intasome structure. We thank the EM and structural biology STP at the Crick for advice, computational and technical support. Histone plasmids were a gift from Joe Landry (Addgene) and 601 sequence was a gift from John Widom (Addgene). We are grateful to Pavel Afonine for help with Phenix real space refinement. This work was funded jointly by the Wellcome Trust, MRC and CRUK at the Francis Crick Institute (FC0010061, FC0010065). A.C. receives funding from the European Research Council (ERC) under the European Union's Horizon 2020 research and innovation programme (grant agreement No 820102). M.D.W. was funded by a Human frontiers Science Program long term Fellowship. The Single Molecule Imaging Group is funded by a core grant of the MRC-London Institute of Medical Sciences (UKRI MC-A658-5TY10), a Wellcome Trust Collaborative Grant (206292/Z/17/Z) and a Leverhulme Grant (RPG-2016-214).

## Author contributions

A.C. and P.C. initiated the study. D.P.M. assembled the Intasome–NCP complex and LR determined the structure. D.P.M. performed pull-down assays. M.D.W. performed biochemistry, assembled NCP-D02-streptavidin and NCP-601 complexes and determined the structures. V.E.P. built all models into the EM maps. M.G. performed single molecule FRET experiments and data analysis, supervised by DSR. M.D.W., L.R. prepared and screened the cryo-EM grids and M.D.W. and A.N. collected cryo-EM data. M.D.W., P.C., and A.C. wrote the manuscript with input from the authors.

## Additional information

**Competing interests:** The authors declare no competing interests.

