## [Peer Review File · Nature Communications]

Reviewers' comments:

Reviewer #1 (Remarks to the Author):

In this work by the Costa, Cherepanov, and Rueda groups, the authors present a 3.9 Å resolution cryo-EM structure of the prototype foamy virus intasome engaged with a nucleosome core particle. The structure, complemented with elegant single-molecule FRET experiments, immediately suggests a detailed mechanism for the structural changes on the nucleosome that are required for retroviral integration: nucleosomal DNA is shifted by approximately two base pairs past the histone core, with a concomitant small DNA loop lifted off the H2A/H2B surface.

This paper follows up on previous structural characterizations of the PVF intasome-nucleosome complex by Costa and Cherepanov. This earlier work had revealed an overall very similar architecture of the nucleosome-bound integration machinery. While it might therefore be argued that the structure presented in this current study is not fundamentally new per se, it is extremely important to stress that the previous structure was determined at substantially lower resolution, which prevented any detailed mechanistic understanding of retroviral integration into the nucleosome. In stark contrast, this current structure allows the authors to put forward a completely new loop-and-sliding mechanism for retroviral integration that is further corroborated by single-molecule FRET analysis from the Rueda lab.

The paper is well written and extremely clear and the experimental design is elegant. Particularly noteworthy is the clever use of a biotin moiety on the distal DNA arm to facilitate asymmetric particle alignment, resulting in an isolated D02 NCP structure of sufficient quality (highly challenging) to be compared to the intasome-NCP complex.

In sum, this well-executed and elegant study represents an important and timely advance that will be of substantial interest to a broad readership in various areas of chromatin and structural biology. I therefore enthusiastically recommend publication of this manuscript, essentially as is, at Nature Communications and only have very minor comments:

1) The authors utilize a FRET-based approach to detect conformational changes upon addition of intasome/Mg. The presence of two H2A copies per histone octamer gives rise to a heterogeneous labeling of the nucleosome with respect to Cy5. It would be good if the authors could include a FRET distribution for the labeled nucleosomes that shows both proximal and distal labeling configurations in the same histogram.

What was the labeling efficiency?

2) Addition of intasome/Mg gives rise to a second, lower-FRET peak, which is appropriately fit with a second Gaussian in Fig. 5F. The fit is not particularly convincing and could potentially be improved by increasing the sample size. I appreciate however that this may not be possible given the close proximity of the centroid positions.

3) Along the same lines, it is not clear why the distributions in C-E were fit with a sum of two Gaussians as in F.

4) The manuscript draws parallels between its loop-and-sliding mechanism and ATP-dependent chromatin remodeling. In light of this, the authors should consider extending their discussion to include recent single-molecule work (Sabantsev et al., Nature Communications 2019) that describes buffering of a small number of base pairs within the nucleosome as a salient feature of ATP-dependent chromatin remodeling, which may also be harnessed by other chromatin-interacting machinery.

Reviewer #2 (Remarks to the Author):

The authors have previously reported the cryo-EM structure of the precatalytic prototype foamy virus intasome in complex with a nucleosome core particle. The most notable feature of this structure was the looping out of DNA on the nucleosome surface at the site of intasome binding. This looping out is required to accommodate the sharp kink that target DNA adopts when engaged with intasome. However, the resolution of the previous structure was insufficient to reveal much detail beyond the lifting of DNA from the surface of the nucleosome at the site of intasome

binding. The new 3.9 Å structure of the postcatalytic complex reveals a much more detailed picture of the structure. As expected, the overall features are similar. The previous structure did not reveal how lifting of DNA from the nucleosome surface was accommodated, but the new structure, together with FRET data, shows that it is achieved by shifting the register of the flanking DNA on the nucleosome surface. It also reveals interactions of the nucleosome with the N-terminal histone H2A tail. This is high quality work that is well presented. However sliding of DNA on the nucleosome surface to accommodate the loop out is not surprising; the alternative of partial unwinding of flanking DNA would be energetically unfavorable. A more specialized journal may be more appropriate for publication of this work.

Minor comment:

Line 260: The suggested explanation for the requirement for magnesium to see a drop in FRET efficiency is unclear; the loop out (and by inference change in DNA register) is present prior to catalysis so why is the change in FRET only observed after catalysis?

Reviewer #3 (Remarks to the Author):

Retroviral integrase efficiently integrates the viral DNA into nucleosomes. A previous low resolution structural study by this collaborative team revealed that integrase directly contacts with the nucleosomal DNA by lifting DNA off the histone octamer surface at its integration site. To study conformational change of the nucleosomal DNA during vDNA integration process, in this study, the authors determined a cryo-EM structure of the prototype foamy virus intasome with a nucleosome at 3.9 Å resolution. The authors then found that the integrase induces two base-pair register shifting of the nucleosomal DNA, accompanying with the DNA lifting from the H2A-H2B surface. The DNA register shifting was also confirmed by the single-molecule Förster resonance energy transfer measurements. This is a high quality cryo-EM study, and interpretation of data is appropriate. Especially, it is an important progress that the authors clearly observed the cleavage site in the nucleosomal DNA. Therefore, I have only minor comments for improvement.

Comments:

The authors performed the structural analysis with Relion1.3 or 2.1. Relion3 may improve resolution better, although the conclusion of this work may not be affected. This is not essential issue.

In Figure S1d, cryo-EM map is missing in Euler angle distribution plot. please add it.

Please indicate scale of 2D class average in Figure 3B, Figure S1b, Figure S3c, and Figure S5c.

Scale bars in Figure S3f and Figure S5d represented as 25 nm. I suppose it should be angstrom. Please correct it.

What is the meaning of a caption “?” in Supplementary Fig2e, lane 5.

In the text, Line 246, I don't understand why the author cites Supplementary Figure 7. Please explain it or change the citation, if it is typo.

In Supplementary Fig6, it is helpful, if the authors show rmsd plots for Ca atoms for comparison between histone octamers, and discuss the relation between the octamer deformation and register shifting of the nucleosomal DNA.

Hitoshi Kurumizaka

We would like to thank the reviewers for comments that helped us improve our manuscript. Below we address the issues raised.

Reviewer 1:

We are thankful to this reviewer for very positive comments on our work. We are happy that this reviewer believes that our study “represents an important and timely advance”. Here is a point-by-point reply to the their comments:

1) The authors utilize a FRET-based approach to detect conformational changes upon addition of intasome/Mg. The presence of two H2A copies per histone octamer gives rise to a heterogenous labeling of the nucleosome with respect to Cy5. It would be good if the authors could include a FRET distribution for the labeled nucleosomes that shows both proximal and distal labeling configurations in the same histogram.

As noted by the reviewer, we observe a mixed population of nucleosomes with tails labelled on either copy of H2A in the octamer. As a consequence, we detect markedly distinct FRET efficiency signals, depending on whether H2A tails proximal or distal from the labelled DNA are measured. As requested by the reviewer, in Supplementary Figure 7 we have added a FRET distribution for the labelled nucleosomes, showing both proximal and distal labelling configurations in the same histogram.

What was the labeling efficiency?

K119C labelling efficiency was 68%, as established by spectrophotometric (A280 vs A650) measurements, and correct labelling was confirmed by intact mass spectrometry. We have now modified the main methods section to include this information. Assuming a normal distribution, we expect the majority of nucleosomes to contain a single label on H2A K119C. Because of the experimental design, non-labelled nucleosomes do not produce a FRET signal. To ensure that nucleosomes labelled on both K119C residues were excluded from our measurements, we only quantified FRET traces where Cy3 was seen to bleach to background levels, during the course of the experiment.

2) Addition of intasome/Mg gives rise to a second, lower-FRET peak, which is appropriately fit with a second Gaussian in Fig. 5F. The fit is not particularly convincing and could potentially be improved by increasing the sample size. I appreciate however that this may not be possible given the close proximity of the centroid positions.

3) Along the same lines, it is not clear why the distributions in C-E were fit with a sum of two Gaussians as in F.

We have done one of the fits with a double and a single Gaussian distribution:

The figure clearly shows that the single distribution fit cannot represent the small shoulder at lower FRET values (smaller, randomly distributed residuals - blue curve). Based on this evidence, we think the double distribution is warranted. The observed shoulder derives from a small fraction of dynamic molecules that transiently visit lower FRET states (see **Sup. Fig. 7**), thereby lowering the average FRET value. This shoulder is different, however, from the large population observed at lower FRET that arises in the presence of IN/Mg²⁺ (**Figure 5F**). With the latter conditions in fact, we reproducibly detect a significant population of molecules featuring a ~0.15 decrease in FRET efficiency. We agree with the reviewer that additional data will not improve the distribution, given the close proximity of the centroid positions. Based on our experience with a very similar experimental design (Wilhoft and Ghoneim *Science*, 2019), we are confident that the change in FRET observed with the addition of IN/Mg²⁺ reports on an alteration in nucleosomal register. For clarity, we would prefer not to change the fit in the figures.

4) The manuscript draws parallels between its loop-and-sliding mechanism and ATP-

dependent chromatin remodeling. In light of this, the authors should consider extending their discussion to include recent single-molecule work (Sabantsev et al., Nature Communications 2019) that describes buffering of a small number of base pairs within the nucleosome as a salient feature of ATP-dependent chromatin remodeling, which may also be harnessed by other chromatin-interacting machinery.

We thank the reviewer for this important remark. We now reference the Sabantsev paper in the main text.

Reviewer 2:

We thank this reviewer for noting that our new 3.9 Å resolution reconstruction of the postcatalytic nucleosome-intasome “reveals a much more detailed picture” of the complex. The reviewer acknowledges that the “new structure, together with FRET data” reveals that nucleosomal DNA looping essential for integration “is achieved by shifting the register of the flanking DNA on the nucleosome surface”. Finally, the reviewer acknowledges our biochemical and structural analysis revealing “interactions of the nucleosome with the N-terminal histone H2A tail”. We would like to thank this reviewer for stating that “this is high quality work that is well presented”.

However sliding of DNA on the nucleosome surface to accommodate the loop out is not surprising; the alternative of partial unwinding of flanking DNA would be energetically unfavorable.

We now state in the introduction that underwinding of flanking DNA could be one alternative to nucleosomal DNA repositioning. DNA binding proteins such as SSB can melt DNA in the absence of any ATPase activity. It is therefore unclear to us why this reviewer states that nucleosomal DNA repositioning should be energetically more favourable than DNA underwinding. Notably, only one case has been reported so far (which was published after our first original submission), where nucleosomal DNA repositioning occurs upon binding of an interactor that lacks ATPase activity. We understand this observation was deemed noteworthy, given that it featured in the title of a recent article on UV-DDB bound to nucleosomes bearing damaged DNA (Matsumoto et al Nature 2019). This study is now cited in our main text. We are likewise excited about our discovery that IN induces nucleosomal DNA repositioning to allow DNA access to the catalytic site for integration.

A more specialized journal may be more approximate for publication of this work.

It should be noted that Reviewer 1 states that our manuscript describes “completely new loop-and-sliding mechanism for retroviral integration”. In Reviewer 3’s opinion, observing a cleavage site in the nucleosomal DNA represents “important progress”. Thus, we respectfully disagree with Reviewer 2 and strongly agree with Reviewers 1 and 3 about the importance of our work.

Minor comment:

Line 260: The suggested explanation for the requirement for magnesium to see a

drop in FRET efficiency is unclear; the loop out (and by inference change in DNA register) is present prior to catalysis so why is the change in FRET only observed after catalysis?

We have expanded on our discussion regarding magnesium requirement to observe a drop in FRET efficiency. As pointed out by the reviewer, according to crystallographic work, the degree of bending required for capturing target DNA matches that required for strand transfer. Likewise, in the pre-assembled intasome-NCP complex, the nucleosomal DNA path pre- and post-integration appears virtually identical. In our single-molecule experiment, however, a change in FRET was only observed in the presence of magnesium that promotes integration. This is expected given that our FRET assay was performed by flowing intasome on a glass slide with tethered nucleosomes, and not using a pre-assembled intasome-NCP complex. Single-molecule work by Fishel and Yoder, performed under comparable experimental conditions (though using non-chromatinised DNA), showed that Intasome spends most of the time scanning DNA by one-dimensional diffusion, before suitable DNA segments are recognised for strand transfer (Jones et al. *Nat comm*, 2016). When magnesium is omitted in our experiment, we therefore expect that intasome will spend most of the time scanning nucleosomal DNA. This search is unlikely to result in any DNA deformation. As the looped & register-shifted state is expected to accumulate upon integration, it is therefore not surprising that the nucleosomal DNA sliding can be clearly detected in the presence of magnesium, which is essential for catalysis.

As suggested by reviewer 1, an alternative explanation is that DNA perturbation and sliding could be functionally distinct steps. This could be mediated by histone buffering of DNA displacement, as observed previously³³. In this model tension in the lifted DNA at the integration site could be accommodated by protein-DNA contacts within the target capture complex, without a concomitant shift in DNA, which only occurs upon full strand transfer catalysed upon addition of magnesium.

Reviewer 3

We thank this reviewer for stating that our study is “high quality” and “appropriate” data interpretation, and that observing the cleavage site in the nucleosomal DNA is deemed “important progress”.

Comments:

The authors performed the structural analysis with Relion1.3 or 2.1. Relion3 may improve resolution better, although the conclusion of this work may not be affected. This is not essential issue.

We agree and attempted reprocessing one of our datasets (Widom 601 NCP) with Relion3 and obtained no improvement, including using new implementations of CTF refinement and Bayesian polishing. Given that all conclusions of our study are solidly supported by the structures presented at the current resolution, and validated by

single-molecule FRET and mutagenesis studies, we believe that reprocessing our data with more recent releases of the same software package is not necessary in this instance.

In Figure S1d, cryo-EM map is missing in Euler angle distribution plot. please add it.

We have now added the Euler plot in Figure S1d.

Please indicate scale of 2D class average in Figure 3B, Figure S1b, Figure S3c, and Figure S5c.

We have now added scale bar for 2D averages in the captions of Figures 3B, S1B, S3C and S5C.

Scale bars in Figure S3f and Figure S5d represented as 25 nm. I suppose it should be angstrom. Please correct it.

We would like to thank the referee for spotting this mistake. This has now been corrected.

What is the meaning of a caption “?” in Supplementary Fig2e, lane 5.

This was a mistake, thanks for spotting it. Lane 5 in Supplementary Figure 2e should be labelled Mg^{2+} . Now corrected.

In the text, Line 246, I don't understand why the author cites Supplementary Figure 7. Please explain it or change the citation, if it is typo.

Thanks for spotting this. Supplementary Figure 7 should not be cited here and has been removed.

In Supplementary Fig6, it is helpful, if the authors show rmsd plots for Ca atoms for comparison between histone octamers, and discuss the relation between the octamer deformation and register shifting of the nucleosomal DNA.

This is a great suggestion. In Supplementary Figure 6, we have now added the RMSD plots for Ca atoms comparing histone octamers in the strand transfer complex and isolated D02 nucleosome. In the Discussion, we now examine the relation between octamer deformation and nucleosomal DNA shifting.